# Clock Frequency Impact on the Performance of High-Security Cryptographic Cipher Suites for Energy-Efficient Resource-Constrained IoT Devices [note 1]

**DOI:** 10.3390/s19010015

**Published:** 2018-12-20

**Authors:** Manuel Suárez-Albela, Paula Fraga-Lamas, Luis Castedo, Tiago M. Fernández-Caramés

**Affiliations:** Department of Computer Engineering, Faculty of Computer Science, Universidade da Coruña, 15071 A Coruña, Spain; m.albela@udc.es (M.S.-A.); luis.castedo@udc.es (L.C.)

**Keywords:** ECC, ECDSA, RSA, IoT, TLS, power consumption, IoT security, energy efficiency

## Abstract

Modern Internet of Things (IoT) systems have to be able to provide high-security levels, but it is difficult to accommodate computationally-intensive cryptographic algorithms on the resource-constrained hardware used to deploy IoT end nodes. Although this scenario brings the opportunity for using advanced security mechanisms such as Transport Layer Security (TLS), several configuration factors impact both the performance and the energy consumption of IoT systems. In this study, two of the most used TLS authentication algorithms (ECDSA and RSA) were compared when executed on a resource-constrained IoT node based on the ESP32 System-on-Chip (SoC), which was tested at different clock frequencies (80, 160 and 240 MHz) when providing different security levels (from 80 to 192 bits). With every tested configuration, energy consumption and average time per transaction were measured. The results show that ECDSA outperforms RSA in all performed tests and that certain software implementations may lead to scenarios where higher security-level alternatives outperform cryptosystems that are theoretically simpler and lighter in terms of energy consumption and data throughput. Moreover, the performed experiments allow for concluding that higher clock frequencies provide better performance in terms of throughput and, in contrast to what may be expected, less energy consumption.

## 1. Introduction

The rise of the Internet of Things (IoT) paradigm brings the opportunity of having any device connected anytime and everywhere. Such a heterogeneity and ubiquity raise some challenges and threats when compared with other more isolated and controlled communications.

Nowadays, IoT devices are connected to the Internet in diverse environments and fields such as industry or defense and public safety [1]. Nevertheless, security issues pose risks for human safety and privacy [2], and it can be stated that the broad adoption of IoT is slowed down due to privacy and security requirements that have not been addressed completely [3].

Although IoT devices have many advantages in terms of scalability and cost, they are restricted in terms of memory, battery, computing capabilities and hardware resources, thus making it difficult to implement the complex and heavy operations needed by ciphering algorithms to encrypt and secure communications [4]. Moreover, some resource-constrained IoT devices might be deployed in large areas where power outlets are not available, so energy harvesting techniques have to be introduced.

Since IoT networks usually rely on TCP/IP, the use of already proven security protocols seems to be the best approach in terms of reliability and implementation efficiency. In this scenario, Transport Layer Security (TLS) arises as the main alternative, but it has a relevant limitation: its most popular cipher suites were not designed for resource-constrained IoT devices [5]. In the last years, UDP-based solutions such as Datagram Transport Layer Security (DTLS) [6] have been introduced to provide lightweight alternatives, but, when security is added to such solutions, their gains over TLS are diminished, being better to make use of broadly available and optimized TLS implementations [7].

One important consideration when securing the communications of resource-constrained devices is the possibility of modifying the hardware configuration to improve both performance and energy consumption. In this aspect, IoT System-on-Chip (SoC) clock frequency is a factor that usually presents a relevant impact, since cryptographic algorithms require high computational capabilities. Moreover, it is possible to modify and test the impact of different SoC frequencies on different hardware platforms with minimum modifications. In contrast, other hardware optimizations such as the modification of the memory mapping [8] are platform-specific and therefore more difficult to implement and test.

This study evaluated the impact of modifying the clock frequency on resource-constraint IoT devices when executing TLS with different cipher suite configurations, thus providing a variety of security levels that ranges from 80 to 192 bits. First, Rivest–Shamir–Adleman (RSA) was evaluated in terms of security, scalability, power consumption and data throughput. The results were then compared with a more suitable approach for resource-constrained devices based on Elliptic Curve Cryptography (ECC). The evaluated ECC key sizes were selected to maintain an acceptable security level for the years to come, unless breakthroughs in ECC cryptography or in certain disruptive technologies (e.g., quantum computing) happen. In fact, IoT node hardware was selected with the intention of being as close as possible in terms of computational power to the nodes that will be deployed in the next generation of IoT networks.

The rest of this article is organized as follows. Section 2 reviews some of the latest academic publications that deal with IoT security energy efficiency. Moreover, it introduces the basics of TLS and its cipher suites, and analyzes the state of the art related to the performance of RSA and ECC when executed by resource-constrained devices. Section 3 details the design of the proposed system. In Section 4, the experimental setup and the performed tests are described. Finally, Section 5 is devoted to the conclusions.

## 2. Related Work

### 2.1. IoT Architectures and Resource-Constrained Devices

In the last years, IoT architectures have evolved towards hierarchical topologies such as the one depicted in Figure 1. Figure 1 shows a generic IoT architecture, with the main elements involved on most IoT deployments. It must be taken into account that IoT devices carry out different tasks depending on their role on the global IoT architecture. Multiple gateways exchange data with the IoT nodes and with the cloud in a hierarchical way. Thus, the cloud at the top of the architecture provides a single point-of-entry for cloud-centric IoT architectures. In comparison, edge computing approaches unburden the higher layers of the architecture and distribute the computational requirements and system access capabilities throughout all the elements involved.

### 2.2. IoT Security Challenges and Limitations

Recently, the number of IoT hardware boards has grown dramatically thanks to the progress made on SoC technologies in terms of energy consumption and computational power, which allows them to provide computer-like capabilities. For instance, in [5], the authors presented a comprehensive analysis of the latest hardware platforms that can be used as IoT gateways and end devices. Among such platforms, Single Board Computers (SBCs) seem to be better suited to act as gateways due to their improved computational performance in comparison to IoT motes and other embedded hardware platforms, which are often designed to operate as IoT end nodes. Nonetheless, most IoT nodes embed low-power computational devices that can perform specific (but basic) tasks and network communications through the Internet, so they are usually considered as resource-constrained devices [9].

Due to the limited capabilities of IoT end devices, the data they capture are transferred to the upper layers of the architecture to process them and provide the required services to users. The direct consequence is that the amount of data per transaction increases as ascending from the bottom to the top of the architecture due to data aggregation from the lower layers. Therefore, it can be observed that the simplest IoT devices in terms of hardware would be at the bottom. However, more power is required as ascending in the IoT architecture, especially for large deployments where thousands of IoT nodes may exchange data simultaneously.

For this reason, the current tendency is to move from pure cloud architectures to edge computing architectures [10,11,12]. Figure 1 presents the main elements of an IoT architecture, comparing the tasks that each of the layers has to perform on a cloud-centric and on an edge computing architecture. As can be observed, most of the computational resources on a cloud architecture have to be concentrated at the cloud layer, since it performs most of the work. Edge computing approaches leverage the potential of the lower layer devices, such as gateways and end devices, to perform either totally or partially the tasks traditionally carried out by the cloud layer. Tasks related to provide IoT services, user access control or to security policies, are performed by the lower layers, thus reducing the computational requirements of the cloud infrastructure. In addition, this distributed approach also eliminates the presence of a single point-of-failure of cloud-centric architectures. Furthermore, thanks to moving the computational resources to the lower layers of the architecture, it is possible to process, persist and provide access to IoT system data closer to where they are produced and consumed, greatly reducing service latency and energy consumption [13].

To allow IoT deployments to move from cloud to edge computing, the devices of the lower layers must be able to perform most of the tasks previously carried out by the upper layers [14]. However, it is important to note that the weakest link in terms of security of an IoT architecture is usually the end device layer, since end devices have to accommodate the computational demands of robust security schemes to their resource-constrained hardware capabilities. In practice, this situation often derives into the fact that IoT developers overlook or address security in a light way, so weak implementations are common, what arises public security concerns [3]. For example, in [9], TLS is used for performing only symmetric key operations, but the proposed cipher suite is currently not recommended since it is considered insecure [15].

Traditional IoT deployments overcome these limitations and the lack of security of the end nodes by centralizing the access permissions and policies on gateways [16,17]. The direct consequence of these implementations is not only an impact on the performance and throughput capabilities of the gateways, but also the introduction of a single point of failure in IoT systems. If a gateway that provides access to a large number of IoT end devices is compromised, all of the IoT nodes that it serves also become compromised, and there is no straightforward mechanism to re-gain trustful access to them.

Moreover, a relevant number of vulnerabilities has been discovered recently after analyzing technologies related to IoT deployments. For instance, in [18], the authors analyzed the security of communications protocols used by Wi-Fi, Bluetooth, RFID or ZigBee. Another interesting survey is presented in [19], which addresses the challenges that arise in terms of security in heterogeneous networks. Finally, another interesting work is presented in [20], where the main security threats that can be exploited to compromise IoT end nodes are studied.

### 2.3. Alternatives for Improving IoT Security

Several authors have proposed different alternatives to tackle the issues that arise when creating secure mechanisms for IoT networks. Some authors have already provided a good list of the security requirements for IoT technologies [21], while other researchers have focused on creating secure IoT architectures [22,23]. Moreover, some researchers have proposed security improvements on the architecture by following a layered approach [24].

Another challenge arises when providing secure end-to-end communications to resource-constrained networks [25]. Thus, the use of protocols such as Datagram TLS (DTLS) has been proposed [6], which was developed for providing security to UDP-based protocols like Constrained Application Protocol (CoAP) and whose overhead can be decreased for using it in sensor networks or constrained devices [26]. Note that since CoAP was also designed for resource-constrained devices; its security is addressed lightly, including the use of pre-shared keys or raw public key cipher suites [27]. Certificates can be used in CoAP, but the supported cipher suites usually provide low security levels.

Finally, it is also worth mentioning a growing tendency in IoT security that is based on the use of unique and unclonable physical structures to avoid some attacks [28].

### 2.4. Security Level

When comparing different cryptographic algorithms, key size (i.e., the number of bits of the key) cannot be used directly as a measure of the strength provided by an algorithm. A better option is to use the security level, which is a value that quantifies the required effort to break a cryptographic primitive [29]. If the effort is $2^{k}$, it is said to offer k-bit security and, therefore, it provides a security level *k*. Different types of cryptographic algorithms (e.g., symmetric, asymmetric, and hash functions) present different relationships between the key size and the security level they provide. For example, considering a k-bit key size, symmetric algorithms provide a k-bit security level while hash algorithms provide k/2-bit security level. Asymmetric key algorithms present different relationships between security level and key size. The National Institute of Standards and Technology (NIST) recommendation for key management [30] provides the reference security levels of symmetric algorithms when compared with two asymmetric algorithms, Elliptic Curve Digital Signature Algorithm (ECDSA) and RSA. These values are summarized in Table 1. For example, a 128-bit security level is achieved by either using 3072-bit RSA or just a 256-bit key size of an ECC curve. For this reason, the concept of security level is used throughout this paper to present a fair comparison among the tested algorithms. The interested reader can find in [31] a comprehensive analysis of the relationship between cryptographic key lengths and security.

### 2.5. TLS for IoT Networks and Cipher Suites

Many IoT systems use TCP/IP communications, whose current best security alternative consists in using TLS [32].

TLS is a standard protocol composed by the TLS Record Protocol and the TLS Handshake Protocol:TLS Record Protocol provides connection privacy and reliability by using symmetric cryptography (e.g., Advanced Encryption Standard (AES) and RC4) and hash functions (e.g., SHA-1).TLS Handshake Protocol enables server and client authentication and helps to determine the encryption algorithm and the cryptographic keys used by the application protocol.

Therefore, the TLS Record Protocol is responsible for securing the connection after the TLS Handshake Protocol establishes the parameters of the TLS session. The procedure is conditioned by the used cipher suite, whose name indicates the involved algorithms. For instance, the cipher suite ECDHE-RSA-AES128-GCM-SHA256 makes use of ECDHE-RSA for the key-exchange, AES128-GCM as block cipher and SHA256 as the hash function that preserves the integrity of the handshake messages.

RSA and Elliptic Curve Diffie–Hellman Ephemeral (ECDHE) are the most popular cipher suites recommended for TLS [15]. The cipher suites based on RSA make use of it for the key-exchange. In contrast, ECDHE-based cipher suites use Ephemeral Diffie–Hellman based on Elliptic Curves for the same purpose.

### 2.6. RSA and ECC Performance Comparisons

Currently, RSA and ECC are not often used by IoT nodes due to their resource demanding requirements, although some researchers proposed resource-efficient hardware implementations of both [33] and evaluated their performance on resource-constrained devices. For example, in [34], the authors evaluated the performance of RSA and ECC on a smart card. Another example is described in [35], where an ECC versus RSA time performance comparison is carried out by using 8-bit CPUs. In this paper, the authors concluded that, for the tested key sizes, ECC outperforms RSA. It is also worth mentioning a report from ATMEL [36] that compares RSA and ECC for embedded systems and concludes that RSA is 10 times slower than ECC for a 128-bit security level while, for a 256-bit security level, ECC is 50–100 times faster.

ECC and RSA have also been studied in terms of power consumption in different scenarios (e.g., [37,38]), but such studies fall short in some aspects (e.g., security levels are not taken into account, the consumed energy is just estimated or outdated hardware platforms and insecure/deprecated cipher suites are used). No paper providing a fair comparison of power consumption in real-world IoT devices with proper security levels, which is presented in Section 4, was found.

### 2.7. Hardware Configuration and Energy Efficiency for IoT Secure End Nodes

There are different approaches when trying to improve the energy efficiency of IoT deployments that target hardware platforms and their configuration. One option is to develop mechanisms that allow for reducing the time end nodes are in an active state [39]. When it is possible to put the nodes to sleep, the reduction in the up-time can yield interesting results on energy consumption, but this is not applicable when real-time constraints or constant availability are required. Some authors investigated energy harvesting techniques [40,41,42], which allow IoT end devices to obtain energy from their environment, and thus remove the need for external power sources and increase battery replacement periods. Other works are focused on improving the network routing protocols of IoT networks to accelerate communications, which has a positive effect on energy consumption [43]. These approximations, although practical in some scenarios, do not address the security mechanisms required to deploy secure IoT networks.

Regarding the modification of the actual hardware configuration parameters of IoT end nodes for improving their energy efficiency, there are few examples in the literature. For instance, the authors of [44] presented an interesting approach that consists of using Dynamic Voltage Scaling (DVS) to improve the energy efficiency of IoT control devices. The proposed scheduling algorithm allows for obtaining significant energy consumption gains while maintaining real-time capabilities. Another example of hardware configuration for improving the energy efficiency of IoT end nodes is presented in [8], where the authors provided energy consumption values for different memory mappings when executed on Ferroelectric RAM (FRAM)-based IoT devices. While both approximations could yield significant energy reductions, they fall short in their applicability: they are complex to implement, platform-dependant and application-specific, narrowing the application of the obtained conclusions to specific hardware platforms and use cases. It is also worth mentioning that the impact of cryptographic algorithms on the energy consumption of IoT end devices can be estimated by means of proper mathematical models [45,46].

### 2.8. Feasibility and Impact of Implementing High Security Mechanisms on IoT End Devices

As explained above, to move from cloud-centric to edge computing IoT implementations, IoT end-devices have to perform tasks previously carried out by the powerful hardware of cloud systems. There are modern hardware boards capable of acting as end-devices, of processing the captured data and of providing complex services, but no actual work was found in the literature that assesses the impact of implementing advanced security mechanisms on such a kind of hardware platforms. There are works evaluating the feasibility and impact of securing IoT communications, but the selected methods do not provide the needed security levels for edge computing deployments. Moreover, although in the literature there are comparisons between different IoT security techniques or algorithms, they are not based on the concept of security level, providing misleading conclusions. Regarding hardware configurations and their impact on securing IoT devices, the works presented in the previous subsections propose complex techniques that cannot be tested on different hardware platforms and whose conclusions are difficult to extrapolate to different systems.

The presented study compared the broadly used RSA algorithm with ECDSA, with keys that provide equivalent security levels. For protecting communications, it uses TLS, one of the currently most used protocols for securing any type of Internet communication. To compare the hardware impact on both tested security schemes, different clock frequencies are tested, since clock frequency is a parameter that can be configured in most hardware platforms and with a presumably high impact on the execution performance of such computational demanding algorithms.

## 3. System Overview

The proposed testbed architecture is presented in Figure 2 and, as can be observed, it is aimed at recording the energy consumption and throughput of an IoT end device. The testbed coordinator is the device in charge of starting, stopping and recording the measurements during the tests. Such a device can be a PC, a laptop or a virtual machine. To power up the IoT end device, a dedicated power supply is used to obtain stable and accurate current measurements. A current sensor is placed between the power supply and the IoT end device to register the current consumed by the end device. The sensor is then connected to an SBC, which is in charge of starting the test procedure when requested by the testbed coordinator. Such a test procedure consists on downloading multiple files from the IoT end device while measuring the energy consumption with the help of the current sensor. To enable the communications among all the computational devices of the testbed, a dedicated communications gateway is used. The gateway is connected to the SBC and the testbed coordinator through a wired connection, while it communicates wirelessly with the IoT end device.

### 3.1. Implemented Testbed

The IoT end device was implemented using an ESP32 module [47] since, to perform the designed tests, an IoT-oriented hardware platform that supports ECDSA and RSA is needed. Moreover, it is required that the SoC of the selected platform allows for using different base clock frequencies. Both requirements are met by the official ESP32 Software Development Kit (SDK), thus making the ESP32 a good alternative against other IoT oriented hardware platforms [14]. The actual IoT board that was used for implementing the testbed was the ESP32-DevKitC [48], which is the official prototyping-oriented version of the ESP32. Such a board integrates all the needed electronic components and a USB connection, easing the development and its integration on final products. The ESP32 embedded on this board provides an IEEE 802.11b/g/n interface and supports Bluetooth 4.2. The core of the SoC is a 32-bit LX6 dual-core microprocessor that operates at up to 240 MHz with 520 KB of SRAM. The hardware acceleration engine for cryptographic algorithms supports AES, SHA-2, RSA, and ECC and also includes a Random Number Generator (RNG). In addition, the ESP32 configuration utility allows for setting three different frequencies when flashing the firmware to the device: 80, 160 and 240 MHz. The module can be powered by a 3.3 V or a 5.0 V power source or by using a micro-USB connector. The selected power supply provides 5 V and a maximum of 2 A, enough for the reduced energy requirements of the ESP32-DevKitC module.

The energy measurement subsystem was formed by an INA219 current sensor and an Orange Pi PC SBC [49] that reads the values reported by the sensor. The INA219 is capable of measuring voltages of up to 32 VDC and currents of up to 3.2 A. However, it can be configured to measure lower voltage and current ranges. For the performed tests, a maximum value of 800 mA was configured, which allows for a resolution of 18 μA. The Orange Pi PC was selected among the different available SBCs because it provides a good compromise between hardware capabilities, energy consumption and cost. With a cost of around $12, it is one of the most affordable SBCs on the market. Moreover, it features an Allwinner H3 SoC, which integrates a Quad-core ARM Cortex A7 that runs at 1.6 GHz. Regarding its communications capabilities, it integrates a Fast Ethernet interface, as well as 40 General Purpose Input/Output (GPIO) pins, which enable the use of the most common standard communications protocols for accessing sensors and actuators (e.g., Inter-Integrated Circuit (I2C), Universal Asynchronous Receiver–Transmitter (UART), and Serial Peripheral Interface (SPI)). These features make the Orange Pi PC ideal for performing the required tasks of the implemented testbed.

Regarding the testbed coordinator, a virtual machine with Debian 9 was used. Such a virtual machine was configured with 4 GB of RAM and four processors. It was executed on a 64-bit Windows 10 PC with 16 GB of RAM and an Intel i7 8550U processor.

To communicate all the wireless components of the testbed, an Asus RT-N12 Wireless N Router was used as communications gateway.

Figure 3 shows a picture of the most relevant elements of the testbed. Specifically, it shows the ESP32-DevKitC (Figure 3A) powered up through the INA219 sensor (Figure 3B), which is connected to the Orange Pi PC (Figure 3C).

### 3.2. Software

The ESP32-DevKitC module was programmed using ESP32-IDF [50]. For each combination of cipher suite, signing algorithm and frequency, a different version of the firmware was uploaded to the ESP32 by using the official flashing tool provided by IDF. Specifically, an Hypertext Transport Protocol Secure (HTTPS) server was implemented for the ESP32 using mbedTLS [51]. The server provides remote users a 512-byte JSON file randomly generated with the Python library Faker v0.73 [52]. The test procedure was started by the Debian 9 virtual machine, which executed a Python script that defined the cipher suite to be used, started the energy measurements using the Application Programming Interface (API) that runs on the Orange Pi PC, and then requested the 512-byte JSON file from the ESP32 a number of times. The time taken by each of the HTTPS transactions was measured and registered. When all transactions were finished, it requested the accumulated energy consumption value from the Orange PI PC and stored all the data into a JSON file. When all the tests were finished, an HTML page used Javascript to parse the collected JSON files and generate charts and tables that show the relevant test data to ease their analysis.

To obtain the current values from the INA219, a Python script was developed. This script first sampled the INA219 sensor by accessing the I2C bus, then collected the obtained values and finally reported the final energy consumption values. Several tests and optimizations to the Python code in charge of sampling the I2C bus were performed, which led to achieving a final sampling rate of 1000 Hz. The script was launched from a Python Hypertext Transport Protocol (HTTP) server that runs on the same Orange Pi PC. The HTTP server provided a REST API, which enables the current measurement procedure to be managed remotely by the testbed coordinator implemented on the Debian 9 virtual machine.

### 3.3. TLS Certificate Selection and Generation

The aim of the proposed tests was to provide a comparative analysis of the performance of RSA and ECDSA in terms of energy efficiency and response time, when securing IoT node communications at different SoC clock frequencies. To provide a fair comparison among the selected algorithms, the tests were driven by the concept of security level. Thus, four different security levels were chosen following the NIST guidelines on TLS implementations, which establish a 112-bit security level as the minimum for public keys. Accordingly, 80-bit (deprecated), 112-bit (minimum), 128-bit (recommended) and 192-bit (future-proof) security levels were selected. Several of the available TLS 1.2 [32] cipher suites that implement RSA and ECDSA signing algorithms were evaluated. To provide valid and applicable results to future standards, the latest TLS standard (TLS 1.3 [53]) was also analyzed. Such a standard establishes the need to implement authenticated encryption algorithms for all the available cipher suites. Thus, two cipher suites that comply with the NIST guidelines and that also provide authenticated encryption (i.e., its block cipher implementation operates in Galois/Counter Mode (GCM)) were selected: ECDHE-RSA-AES256-GCM-SHA384 and ECDHE-ECDSA-AES256-GCM-SHA384. These two cipher suites only differ on their signing algorithm: the former uses RSA and the latter ECDSA. The rest of the algorithms are exactly the same.

Seven different certificates that provide the previously mentioned security levels were generated. For RSA, three key sizes were used (1024, 2048 and 3072 bits), while, for ECC, the certificates were generated using four curves (prime192v1, secp224r1, secp256r1 and secp384r1). In the case of RSA, no certificate was generated for providing a 192-bit security level, since it would require using a 7680-bit key size, which cannot be handled by the ESP32 crypto engine (it only supports up to 4096-bit RSA operations) and, currently, it would make no sense to use it in IoT applications.

## 4. Experiments

The experiments were designed with two major goals in mind: to provide valid conclusions for current and mid-term IoT deployments, and to be easily contrasted and compared with other hardware and software implementations. Specifically, the experiments were designed with the aim of providing insightful results about three different aspects: (1) to test the feasibility of using high security communications mechanisms on the lower layers of the IoT architecture (i.e., the end-devices); (2) to determine whether the use of different ECC curves may yield energy consumption and data throughput improvements over RSA when comparing implementations that provide the same security level; and (3) to determine the impact of modifying the clock frequency of the SoC when executing the computationally-intensive algorithms required to provide the selected security levels.

With the mentioned goals in mind, the ESP32 module was tested for the 21 scenarios that result as a combination of the seven generated certificates and the three available SoC frequencies. For each scenario, the 512-byte JSON file provided by the ESP32 HTTPS server was downloaded 100 times by the virtual machine, while obtaining different measurements of energy consumption and throughput.

### 4.1. Initial Setup

Before performing the experiments, the ESP32-DevKitC module was programmed with each of the selected configurations to verify the proper functioning of the generated certificates and the involved cipher suites. The ESP32 was configured for all the tests with Secure Hash Algorithm (SHA) and AES hardware acceleration enabled. NIST modulo *p* optimizations for ECC were used in all tests because they improve the performance on both cipher suites (during the ECDHE key exchange phase, ECC operations are performed). For RSA certificates with a key size of 1024 and 2048 bits, hardware acceleration for Multi-Precision-Integer (MPI) was used. For the 3072-bit RSA certificate, it was not possible to use MPI hardware acceleration: when such an acceleration was enabled, the ESP32 module rebooted as soon as the HTTPS connection was initialized. Considering that the ESP32 datasheet [54] states that the crypto engine supports up to 4096-bit operations for RSA, the problem seems to be software related. With the ECC certificates, no further issues were found.

### 4.2. Energy Consumption Results

Energy consumption was obtained for each of the previously described test scenarios. Figure 4 and Figure 5 show, respectively, the obtained results for ECDHE-RSA-AES256-GCM-SHA384 and ECDHE-ECDSA-AES256-GCM-SHA384 cipher suites. The x-axis represents the SoC frequency. For each of the frequencies, the security level increases from the left to the right. The actual RSA key size and used ECC curve is depicted on the legend of the figure. The bars in the same color correspond to security levels that are equivalent for RSA and ECC.

To ease the comparison, Table 2 presents the energy reductions obtained when using the tested ECDSA and RSA cipher suites for different security levels and SoC frequencies. As can be observed, RSA is outperformed by ECDSA on every scenario. Even though the differences between RSA and ECDSA cipher suites decrease as the frequency increases, ECDSA consumes less energy than RSA for all tested configurations of SoC frequency and security level. In particular, when configured at a frequency of 240 MHz (the scenario in which less differences are obtained), ECDSA reduces RSA energy consumption by a 35.75% for an 80-bit security level, a 22.65% for a 112-bit security level and a remarkable 65.59% for a 128-bit security level.

Regarding the impact of the frequency on the reported energy consumption, it can be concluded that the use of the highest available SoC frequency provides the bests results in terms of energy efficiency.

When comparing only the energy consumption of the ECDSA cipher suite, it can be seen that the curve secp224r1 performs worse than the curve secp256r1 when the SoC is configured to run at the maximum frequency. This because the NIST modulo *p* optimizations are platform and curve-dependent, which means that, even if a curve uses fewer bits (and as a consequence provides a lower security level), it can be outperformed (in terms of energy efficiency) by a more secure curve. Since such optimizations are based on software implementations, increasing SoC frequency will have a greater impact on the curves whose implementations are optimized. Thus, the relative performance between these two curves varies with the frequency. Specifically, it can be observed that at 80 MHz secp256r1 performs worst, at 160 MHz both curves present similar energy consumption and at 240 MHz the secp224r1 is outperformed by the more secure and more optimized secp256r1 curve.

To verify that the secp256r1 curve actually outperforms secp224r1 due to the previously mentioned platform-specific optimizations, the same tests were performed at 240 MHz but disabling the NIST modulo *p* optimizations. The results are shown in Figure 6. As expected, the less secure secp224r1 curve presents lower energy consumption values than the secp256r1 curve, which confirms the hypothesis and thus illustrates the impact of the NIST modulo *p* optimizations.

### 4.3. Throughput Results

Regarding throughput, the conclusions that can be drawn are very similar to the ones obtained for energy consumption in Section 4.2. Table 3 show the average time required per each of the 100 HTTPS requests performed for the different RSA and ECDSA configurations. As can be observed, ECDSA outperforms RSA in all scenarios. Specifically, when running at 240 MHz and for a 128-bit security level, the secp256r1 is roughly three times faster than 3072-bit RSA. Moreover, the secp384r1, which provides a 192-bit security level, is as fast as the weaker 2048-bit RSA that provides only 112 bits of security. For each of the tested configurations, the fastest result is achieved when configuring the SoC at 240 MHz, which provides more than a 50% time per request reduction when compared to running the SoC at 80 MHz. The only exception occurs when using the secp224r1 curve, where a 46.2% reduction in time per request is achieved. This fact is related to the observed behavior of the secp224r1 curve regarding the NIST modulo *p* optimizations. In fact, when comparing the throughput performance of the secp224r1 curve with one obtained by the secp256r1 curve when the SoC runs at 240 MHz, the latter outperforms the former, presenting a time per request reduction of over 10%.

### 4.4. Comparative Analysis of ECDSA and RSA Cipher Suites for the Obtained Results

The obtained results show huge differences for the same security level when comparing ECC and RSA, being ECC a better alternative, since it presents less energy consumption and higher data throughput values. However, the results also present interesting findings when different implementations and key sizes of the same algorithm are compared.

Regarding ECC curves, the most interesting finding is the fact that the security level provided by a curve is not always proportional to its performance. With the ESP32-IDF version used for the tests in this article, the secp256r1 curve presented lower energy consumption and higher throughput values than the weakest secp224r1 curve when the SoC is running at 240 MHz. This fact is explained because the software optimizations performed to speed up the mathematical operations of the algorithms involved in the curves are platform and curve dependent. This fact was empirically demonstrated by repeating the same test with the NIST modulo *p* optimizations disabled, causing the secp256r1 curve to consume more energy than the secp224r1 curve.

With respect to the RSA key sizes, the main problem is that, for a 3072-bit key size, it was not possible to use the hardware acceleration for Multi-precision integer (MPI) operations, which would have a great impact on speeding up RSA calculations and, consequently, reducing energy consumption.

When comparing the general performance of the two tested ciphers suites, it can be concluded that, for the same security level, ECDHE-ECDSA-AES256-GCM-SHA384 always outperforms ECDHE-RSA-AES256-GCM-SHA384 in both energy consumption and throughput.

When taking into account the SoC frequency, another interesting behavior was discovered: increasing the frequency, instead of increasing the energy consumption (as it may be expected), actually improved energy efficiency in all tested configurations. Similarly, as discussed in Section 4.3, it was observed that the SoC frequency also has a major impact on the time per request required when carrying out HTTPS communications. Therefore, the increase in energy consumption caused by using a higher frequency is compensated with shorter communications times, resulting in a reduction of the total energy consumption. Moreover, the obtained results show the impact of the ECC NIST modulo *p* optimizations on different curves and confirmed the platform and curve dependency of this type of optimizations. Specifically, when running the SoC at 80 MHz, the secp256r1 performed worse than the secp224r1 curve, but when rising the frequency to 240 MHZ the secp256r1 outperformed the weakest secp224r1. This fact, along with the test performed with the NIST modulo *p* optimizations disabled, confirmed that the observed behavior was produced by these optimizations.

## 5. Conclusions

The aims of this study were to compare the performance of ECDSA and RSA TLS cipher suites and evaluate the energy consumption impact of the different ECC curves and RSA key sizes when using a resource-constrained IoT node capable of running at different clock frequencies.

After analyzing all the tests, it can be concluded that, in the selected scenarios, ECDSA can be presented as a greener alternative than RSA for securing resource-constrained IoT devices. Regarding clock frequency, the selected hardware platform presented better results in terms of energy efficiency and response time when using the highest available frequency. By testing different working frequencies, it was possible to show the impact of software optimizations that can improve individual ECC curve performance.

The obtained results also emphasize the importance of testing and measuring empirically the performance of the different algorithms supported by hardware platforms. The created testbed allowed accurately comparing the different cipher suites and configurations in terms of security, energy consumption and throughput. For instance, the impact of certain software implementations and optimizations was demonstrated, which can make weaker security alternatives perform worse in terms of energy consumption than more secure approaches. To sum up, real-world scenario testing is a good tool for accurately finding which security algorithm and configuration is the best fitted for an IoT application.

## Figures and Tables

**Figure 1 sensors-19-00015-f001:**
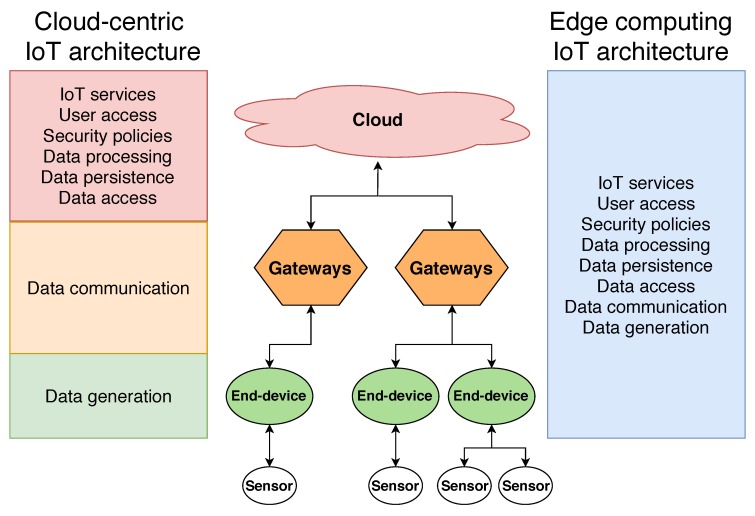
Cloud-centric IoT architecture vs. edge computing IoT architecture.

**Figure 2 sensors-19-00015-f002:**
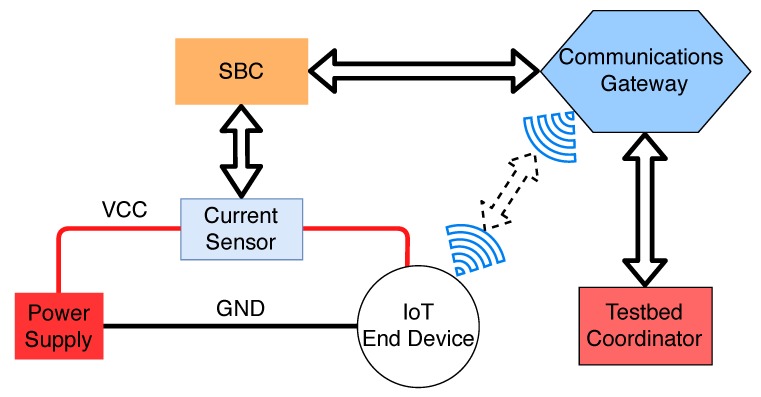
General testbed architecture.

**Figure 3 sensors-19-00015-f003:**
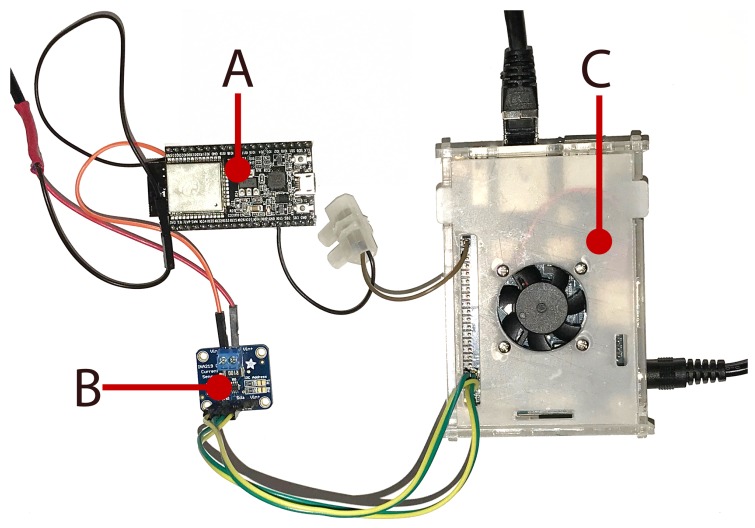
Implemented energy measurement testbed.: (**A**) ESP32-DevKitC; (**B**) INA219; and (**C**) Orange Pi PC.

**Figure 4 sensors-19-00015-f004:**
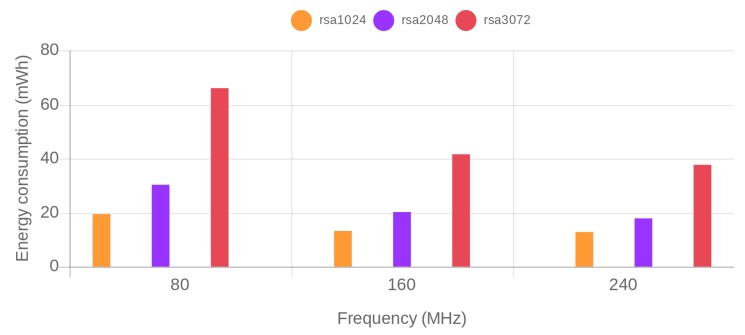
Energy consumption grouped by SoC frequency for the different tested RSA key sizes.

**Figure 5 sensors-19-00015-f005:**
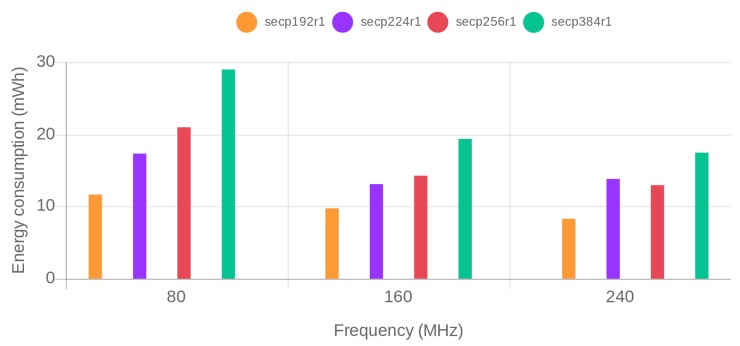
Energy consumption grouped by SoC frequency for the different tested ECDSA curves.

**Figure 6 sensors-19-00015-f006:**
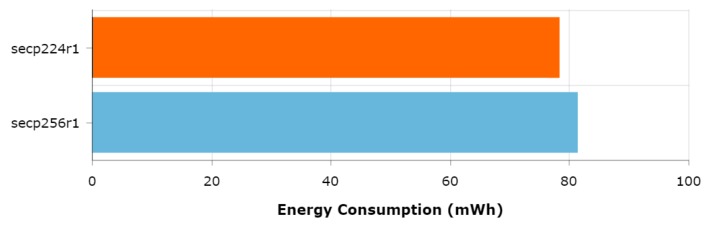
Energy consumption of secp224r1 and secp256r1 curves without NIST modulo *p* optimizations (ESP32@240 MHz).

**Table 1 sensors-19-00015-t001:** Comparable strengths for symmetric, RSA and ECDSA ciphers [30].

Security	Symmetric	RSA	ECDSA
Level	Key Algorithms	Key Size	Curve
80	2TDEA	1024 bits	prime192v1
112	3TDEA	2048 bits	secp224r1
128	AES-128	3072 bits	secp256r1
192	AES-192	7680 bits	secp384r1

**Table 2 sensors-19-00015-t002:** Energy consumption reduction (in percentage) of ECDSA in comparison to RSA.

Frequency (MHz)	80-bit	112-bit	128-bit
80	39.64	42.76	68.21
160	26.36	35.30	65.64
240	35.75	22.65	65.59

**Table 3 sensors-19-00015-t003:** Average time per request (seconds) for the tested RSA key sizes, ECC curves and SoC frequency combinations.

	RSA Key Sizes			ECC Curves			
Frequency (MHz)	RSA 1024	RSA 2048	RSA 3072	prime192v1	secp224r1	secp256r1	secp384r1
80	2.11	3.33	7.51	1.21	1.84	2.31	3.23
160	1.16	1.86	3.84	0.83	1.16	1.27	1.75
240	0.89	1.21	2.63	0.55	0.99	0.88	1.20

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
