# Peer review of "Clock Frequency Impact on the Performance of High-Security Cryptographic Cipher Suites for Energy-Efficient Resource-Constrained IoT Devices†"

_sensors, 2018, doi:10.3390/s19010015_

Reviewer 1 Report

This paper presents a comparative study of implementations of two famous authentication algorithms (ECDSA and RSA) on IoT hardware (ESP32). Through a detailed experimental testbed, this work provides interesting and objective results in terms of energy consumption and thoughput (even if those results are quite expected regarding key size of the two algorithms).

After double checking, this paper is different from https://www.mdpi.com/1424-8220/18/11/3868 despite its relative proximity in time and content.

However, serious flaws can be address to the authors if we consider auto-citation. I can accept one or two autocites but not 12 as it is. Please reduce drastically auto-citation in the final version.

The rest is minor.

1) Compress the abstract (an half is relevant from my point)

2) you should better assume what type of IoT you address. I don't think that your testbed work for LoRa (also natively included in ESP32). if you assume TCP/IP you are already in a particular case of IoT. please, have a word about that

line 136. cyrpto. please use spell checking.

3) Figure 1. what diff btwn net A and B. Authors should change a bit this figure in their numerous articles...

4) I don't agree Table 1: 2TDEA is for me 112 bits, 3TDEA is 168 bits security level...

line 282. [38] is not relevant. [21] yes with precision of consulted pages.

line 297 a key of 7680 has no sense for me for IoT (and for any application) currently.

5) table 3 and 4 have to be merged for a better reading and comparison

time spent for review: 2 hours

Author Response

Dear Sir/Madam,

Please find attached our detailed responses to the comments. 

Regards.

Reviewer 2 Report

In this work the authors deal with the problem of assessing the clock frequency impact on the performance of high-security cryptographic ciphersuites for energy-efficient resource constrained IoT devices.

- The main problem is that the aims of the paper have not been defined or clarified, mainly with respect to the state of the art. In fact, as it is, this paper seems to be little more than a list of measurements, conducted on devices that are the basis of a typical IoT network. Therefore, authors should highlight the benefits of their approach and how, the proposed contributions, impact the lifetime of a secure IoT network infrastructure.

- The authors should clarify what are the aims of the testing phase and whether such aims have been met by the results obtained.

- In some parts of the paper, the clarity and editorial quality of the paper weaken. As a consequence, such parts result to be quite difficult to read. Therefore, I would suggest to carefully improve the prose of writing in order to make this paper easier to read.

- For what concerns the impact of cryptographic algorithms on the energy consumption of network nodes, the authors should mention, by taking into account the following related works, that such a consumption can be estimated by means of proper mathematical model:

https://doi.org/10.1016/j.jcss.2014.12.022

https://doi.org/10.1109/TMC.2006.16

- A thorough proofreading should be carried out, since in the paper there are some typos and formatting issues.

Author Response

Dear Sir/Madam,

Please find attached our detailed responses to the comments. 

Regards.

Round  2

Reviewer 2 Report

p.p1 {margin: 0.0px 0.0px 0.0px 0.0px; font: 15.0px Arial}

The authors addressed all the issues I pointed out, by carefully revising the paper. The paper has been substantially improved with respect to the relative previous version. In my opinion, the paper is now ready to be accepted for publication.